# Hypermethylated Colorectal Cancer Tumours Present a Myc-Driven Hypermetabolism with a One-Carbon Signature Associated with Worsen Prognosis

**DOI:** 10.3390/biomedicines12030590

**Published:** 2024-03-06

**Authors:** Christophe Desterke, Fanny Jaulin, Emmanuel Dornier

**Affiliations:** 1Inserm U1310, Faculté de Médecine, Université Paris Saclay, F-94800 Villejuif, France; christophe.desterke@inserm.fr; 2Inserm U1279, Gustave Roussy, F-94805 Villejuif, France; fanny.jaulin@gustaveroussy.fr; 3Inserm U1016, Institut Cochin, F-75014 Paris, France

**Keywords:** Myc, metabolism, colorectal cancer, CIMP, immunotherapy, one-carbon metabolism

## Abstract

Colorectal cancer (CRC) is the second cause of cancer-related death; the CpG-island methylation pathway (CIMP) is associated with KRAS/BRAF mutations, two oncogenes rewiring cell metabolism, worse prognosis, and resistance to classical chemotherapies. Despite this, the question of a possible metabolic rewiring in CIMPs has never been investigated. Here, we analyse whether metabolic dysregulations are associated with tumour methylation by evaluating the transcriptome of CRC tumours. CIMP-high patients were found to present a hypermetabolism, activating mainly carbohydrates, folates, sphingolipids, and arachidonic acid metabolic pathways. A third of these genes had epigenetic targets of Myc in their proximal promoter, activating carboxylic acid, tetrahydrofolate interconversion, nucleobase, and oxoacid metabolisms. In the Myc signature, the expression of GAPDH, TYMS, DHFR, and TK1 was enough to predict methylation levels, microsatellite instability (MSI), and mutations in the mismatch repair (MMR) machinery, which are strong indicators of responsiveness to immunotherapies. Finally, we discovered that CIMP tumours harboured an increase in genes involved in the one-carbon metabolism, a pathway critical to providing nucleotides for cancer growth and methyl donors for DNA methylation, which is associated with worse prognosis and tumour hypermethylation. Transcriptomics could hence become a tool to help clinicians stratify their patients better.

## 1. Introduction

Colorectal cancer (CRC) is the second cause of cancer-related death worldwide, responsible for nearly a million deaths in 2020 [1]. CRC tumours can arise from two main mechanisms of carcinogenesis: the conventional pathway, associated with mutations in APC and p53, and the serrated pathway, which is associated with mutations in KRAS or BRAF and a hypermethylated phenotype (CpG island methylator phenotype, or CIMP). Serrated tumours, although less frequent, have been shown to be the most aggressive subtype of CRC [2]. Nevertheless, the strong heterogeneity in their pathological presentation and mutational pattern makes them very hard to diagnose. Moreover, multiple methylation panels are used to diagnose the CIMP phenotype, leading to heterogeneity in the diagnosis, and new means of identifying these higher-risk tumours are needed.

The CIMP subtype of CRC is found in around 20% of all CRC patients and is associated with worse survival rates and resistance to classical treatments like 5-Fluorouracil (5-FU) [3]. Moreover, several meta-analyses have highlighted the disparities in the diagnosis of CIMP patients: there is no consensus on the gene panels to be used or in the experimental method of choice to test gene methylation, leading to 16 different definitions of CIMP in the literature [4]. There is hence a need to homogenize our definition of CIMP and find reliable phenotypic markers for this group of patients.

Since KRAS and BRAF are known regulators of metabolism in multiple cancers, driving rewiring in tumour cells, CIMP tumours could harbour a particular metabolic signature, differentiating them from other types of CRCs. Indeed, it has previously been shown that the CIMP phenotype is associated with the up-regulation of enzymes in glycolysis and related processes [5]. Moreover, since CIMP tumours are characterized by a hypermethylation phenotype, it is very likely that the metabolism of these tumours also needs to adapt and accommodate to the higher demand for methyl groups.

Methylation is one of the main epigenetic ways to regulate gene expression; it consists of the addition of a methyl group to the cysteine in position 5 of histones or a direct DNA methylation on cytosines present in CpG islands found in gene promoters [6]. Either methylation on histones or DNA results in the downregulation of the methylated gene and the CIMP phenotype arises from increased DNA methylation. The one-carbon (1-C) metabolism is the central provider of methyl groups needed for methylation processes, and its activation has been shown to be associated with poorer prognosis in colorectal and other cancers [7,8,9,10,11,12]. Indeed, 5-FU, an anti-metabolite chemotherapy targeting 1-C metabolism, is the first line of treatment for CRC patients [13] and one of the most widely used types of chemotherapy [14], indicating the importance of this pathway for cancer cell growth in many different types of cancers. Finally, it has recently been shown that the mitochondrial part of the 1-C metabolism is not required for cancer growth but fosters the migration of cancer cells [15]. Altogether, these data suggest that the 1-C metabolism is important for both cancer growth and dissemination. Nevertheless, its contribution to the CIMP phenotype specifically and its value as a prognostic factor have never been investigated. Furthermore, targeting of the 1-C metabolism pathway is difficult because it is so essential to normal cells and hence generates high levels of resistance to treatment. To date, most of the targeting is achieved through Dihydrofolate reductase (DHFR) or the Thymidylate synthase (TYMS), but more targets for therapeutic intervention are required.

The aim of the present study is to evaluate if CIMP tumours present a distinct metabolic signature, if their levels of methylation could be predicted using transcriptomics data, and if new therapeutic targets can be identified.

## 2. Materials and Methods

### 2.1. TCGA RNA-Sequencing Dataset

The TCGA consortium RNA-sequencing data matrices of Z-scores from the CRC-2012 cohort [16] were downloaded with their corresponding clinical data from the Cbioportal website [17]. Supplemental biological information was also provided for these patients, such as methylation group, MLH1, hypermutation, and microsatellite instability (MSI) status. This cohort was composed of 223 patients, and stratification according to methylation status showed 80 patients with highly methylated tumours (CIMP low and CIMP high) versus 143 patients with unmethylated tumours (cluster3 and cluster4) (Table 1).

The main clinical parameters provided in this study, such as the primary site of tumour localization, oncotree code, and cancer type were found to be significant in this methylation stratification (Table 1). Metabolism pathway enrichment was performed with Geneset enrichment analysis software, version 4.0.1 [18], with KEGG and REACTOME databases implemented in MSigDB, version 7.0 [19]. Bioinformatic analyses were performed in the R-software environment using version 3.5.3. Unsupervised principal component analysis was performed with Factominer version 2.10 R-package [20]. Boxplots were drawn with ggplot2 version 3.5.0 graph definition [21], and expression heatmaps were created with pheatmap version 1.0.12 R-package with the option of Euclidean distances. Machine learning random forest was used for selected transcriptome features during unsupervised analysis. This learning was performed with Randomforest version 4.7-1.1 R-package [22], and the model was built with 150 trees and optimized using the mtry parameter after tuning with the rftune function. Variable importance was kept in the model to interpret gene priority in terms of accuracy.

### 2.2. Overall Survival Analysis Using TCGA RNA-Sequencing with Multi-Omics Integration in Colorectal Cancer

Using the Cbioportal web application [17], RNA-seq data from the TCGA 2018 colorectal cancer cohort (our validation cohort) was analysed with Z-score diploid V2 matrix normalization. This dataset, composed of 379 patients, also had overall survival information [23]. This information was used to investigate one-carbon metabolism gene candidates, as defined by the AMIGO web database of gene ontology information [24] and specified in Appendix A, on patient survival.

### 2.3. ChIP-Sequencing Analysis

Results from ChIP-sequencing experiments performed on the colorectal cell line LoVo [25] were downloaded on the Cistrome project website. Gene promoter annotation of MYC epigenetics intervals was performed with the BETA minus algorithm version 1.0.7 on the human genome, version HG38. Proximal peaks were filtrated around the transcription starting sites 5 kb upstream and 0.5 kb downstream. Functional enrichment on MYC epigenetic genomic intervals (HG38) was predicted using the GREAT website application with the Gene Ontology Biological Process database [26]. The functional enrichment network of GREAT metabolism connections was drawn with Cytoscape software, version 3.6.0 [27]. Motif prediction was performed with RSAT genomic application on human HG38 with the Jaspar vertebrae 2018 motif collection [28]. Promoter heatmap analysis was performed with deeptools, version 3.3.1, in the Linux MINT 19 operating system with custom wrapper in BASH script (https://github.com/cdesterke/chip2heat (accessed on 30 November 2021)). The full results are available in Appendix A.

### 2.4. Integrative Analysis

In order to integrate Myc epigenetics analyses with the RNA-sequencing transcriptome of CRC patients, genomic coordinates from the CHIP-sequencing data were initially aligned on the HG38 human transcriptome with liftover to HG19 genomic coordinates. This operation was done with the liftover algorithm on the UCSC website (https://genome.ucsc.edu/cgi-bin/hgLiftOver (accessed on 30 November 2021)). Circosplot was drawn with the OmicCircos R bioconductor package version 3.18 for multi-omics integration [29].

### 2.5. Deep Learning

Keras version 2.7.0, Scikit learn version 1.0.1, tensorflow 2.7.0, pandas 1.3.5, and matplotlib version 3.5.1 python libraries (python version 3.7.6) were used in the Jupyter notebook version 6.4.6 to implement the deep-learning neuronal network to validate the MYC targets predicting the following clinical data: methylation status, hypermutated phenotype, MLH1 silencing, and MSI status. Neuronal networks were built with three sequential neuron dense layers: the first one with 12 neurons and the Rectified Linear Unit (ReLU) activation function, the second one with 8 neurons and the ReLU activation function, and the third one with one neuron and the sigmoid activation function. The model was compiled with the adam optimizer and loss function based on binary cross entropy and fit with 150 epochs and a batch size of 10. The corresponding python code was deposited at the following web address: https://condescending-saha-2dfa16.netlify.app/ (accessed on 20 December 2021).

### 2.6. Multivariable Model Built on Methylation Status Outcome

In order to test dependencies between qualitative variables of the study, the chi2loop custom designed R-package (https://github.com/cdesterke/chi2loop, accessed on 30 November 2021) was developed to perform iteration chi-square tests between character variables from the cohort dataset. This package is available at the following address: https://github.com/cdesterke/chi2loop (accessed on 30 November 2021). This package takes as input a list of qualitative variables. To run iterative chi-square tests with the cltest function, the dataset needs to be imported with the parameter “stringsAsFactors = FALSE” because qualitative variables as inputs need to be in character format rather than being factors. The cltest function can be applied on the dataset to perform chi-square iteration between character columns, and outputted results can be graphically represented with the nlpplot function (NLP: negative log10 of chi-square test *p*-values). Subsequently, the chinet function can be applied also to the cltest results to detect variable communities with a Louvain classification algorithm. After selection of associated clinical and expression markers, a multivariable model was built with a generalized linear model (GLM) R base function for a logistic regression using a binomial family as parameters (the methylation status was used as the binary outcome).

Graphical output of the multivariable logistic regression model censored according to the binomial status of methylation was drawn with the following R-package combination: broom version 0.7.10, broom.helpers version 1.4.0, and GGally version 2.1.2. In the multivariable logistic model, a *p*-value < 0.05 for included variables in the model was taken as the threshold of significance for their independence.

### 2.7. Statistical Analyses

Statistical analyses were performed in the R software environment, version 4.1.0. For the Z-scores from the RNA-seq data (Gaussian transformation), multi-group comparison was performed with one-way ANOVA Fisher statistical tests, and two group comparison was performed with the bilateral *t*-test with Welch correction. For both approaches, statistical significance was assessed for a *p*-value lower than 0.05.

## 3. Results

### 3.1. Hypermetabolism in CIMP CRC Transcriptome

To investigate the metabolic signature of CIMP tumours, we used the CRC-2012 TCGA cohort [16], comprising 223 patients, where methylation levels were measured using the Illumina Infinium DNA methylation platform (HumanMethylation27 BeadChip technology), and the CIMP status was hence determined on the basis of whole genome methylation and not just a panel of genes. Patients were divided into unmethylated (clusters C3 and C4) versus CIMP patients (clusters CIMP_low and CIMP_high). REACTOME and KEGG databases were used to perform Geneset enrichment analysis (GSEA) to compare the transcriptome of methylated and unmethylated tumours, with an emphasis on metabolic pathways. Surprisingly, we found that all metabolic pathways significantly modulated in CIMP patients were up-regulated. This hypermetabolism associated with the CIMP phenotype comprised 169 enzymes (Appendix A) involved in pathways of the biochemistry of carbohydrates, nucleotides, and sphingolipids (Figure 1A).

We then performed an unsupervised principal component analysis (PCA) using this metabolic signature, which confirmed its ability to separate methylated and unmethylated tumours (*p*-value = 3.35 × 10^−19^, Figure 1B and Appendix A) but was also able to show a progressive stratification of methylation status from negative (clusters c3 and c4) to CIMP_low and CIMP_high on the first principal axis of the unsupervised analysis (*p*-value = 6.48 × 10^−25^, Figure 1C). These results suggest a close association between the CIMP phenotype and this hypermetabolism in CRC patients. Indeed, feeding our 169 metabolic markers into a supervised machine learning algorithm led to discrimination of methylated and unmethylated tumours with a global efficiency of 77.58% after 150 trees of random forest learning (Figure 1D). This efficiency was confirmed by unsupervised classification, performed with Euclidean distances, showing on the expression heatmap a smaller cluster comprising a majority of CIMP tumours and a larger cluster comprising a majority of unmethylated tumours (Figure 1E). A random forest variable importance study further confirmed the importance of glycolysis, nucleotide, and one-carbon metabolism to predict CIMP status, with GAPDH being the best metabolic marker (Figure 1F, Appendix A).

### 3.2. Myc Regulates One-Third of the CIMP-CRC Metabolic Program

We next wondered if there could be a master regulator orchestrating this hypermetabolism, characteristic of CIMP tumours. Myc is a strong promoter of metabolism, fostering cell proliferation and cell fitness [30] but also controlling stem cell fate decisions [31]. It was shown that its expression is changed in nearly all colorectal tumours, suggesting a strong role in CRC pathogenesis [16]. We analysed whether the metabolic rewiring observed in CIMP could be driven by Myc, using publicly available ChIP-seq data from the colorectal cell line LoVo [25]. The promoter heatmap of the 38,663 MYC peaks confirmed that the signal is well centred on the Transcription Starting Sites (TSSs) (Figure 2A). A phast conservation study for Myc confirmed that its chromatin binding events were found to be well conserved in the core mammalian promoter database (Figure 2B). Myc chromatin bindings were then mapped on HG38 human genome promoters, and proximal events were filtered around the TSS (5 kb upstream and 0.5 kb downstream). The resulting 5000 gene promoter prediction was crossed, in the LoVo cell line, with genes from the CIMP metabolic signature; we found 53 genes with Myc binding sites in their promoters (Figure 2C, Appendix A). These results indicate that at least a third of the metabolic signature observed in CIMP patients could be driven by Myc. A genomic Circosplot on the Myc signature revealed that sexual chromosomes, chromosomes 6, 13, 16, and 20, were not involved in the CIMP-specific metabolic program (Figure 2D). Chromosome 12 contained the most genes from the signature, with eight Myc targets, including GAPDH, which was found to be the best predictive marker of the CIMP phenotype (Figure 1F and Figure 2D).

### 3.3. Genes from the Myc Transcriptional Program also Have Binding Sites for Other Transcription Factors

The observed mean size of Myc chromatin binding peaks in the CIMP-specific metabolic program was 651 pb (SD: 259 pb, n = 54 peaks for 53 genes), and the identified peaks were narrow and centred on the Transcription Starting Sites (TSSs) (Figure 3A). This proximal transcriptional program was compatible for performing motif prediction with the RSAT web application, using the JASPAR core promoter motif database. After picking corresponding nucleotide sequences on the HG38 human genome, we performed a transcription factor binding motif identification and confirmed MAX binding sites together with other transcription factors (Figure 3B). Interestingly, we also found 182 compatible sites for homeobox transcription factors like POU6F1, DLX1, and EMX2–142, 156 binding sites for Kruppel-like factors KLF16 and KLF5, and 111, 73, and 78 binding sites for HOX family homeobox transcription factors HOXA13, HOXA1913, and HOXB13 respectively (Figure 3B). We confirmed physical recruitment of DLX1, KLF5, and HOXA13 using the LoVo ChIPseq data (Figure 3C). We also found that the expression levels of MAX, DLX1, and SP8 transcription factors were being progressively overexpressed with the level of methylation, whereas MYC was stable in all subtypes except in CIMP_high, where it was the lowest. Altogether, these results suggested that the Myc metabolic program found in CIMP CRC tumours could be active in a chromatin context implicating homebox and Kruppel-like factors.

Functional enrichment performed on the genes from the Myc signature using the Gene Ontology Biological Process database (GO-BP) showed that the main altered pathways were the carboxylic acid, tetrahydrofolate interconversion, nucleobase containing small molecule, and oxoacid pathways (Figure 4A). This metabolism enrichment highlighted enzymes at the interface between glycolysis (GAPDH, ENO1, LDHA) and one-carbon metabolism (TYMS, SHMT2, MTHFR, MTHFD1), further confirming the importance of these two pathways in CIMP tumours (Figure 4B).

### 3.4. Metabolism Targets in the Myc Signature Are Associated with Worst Clinical Group in CRC

The following clinical parameters are available in the CRC-2012 cohort: methylation status, expression subtype (clustering into three subtypes according to their mRNA profile), hypermutation, and MLH1 silencing [16]. We next evaluated the correlation between these parameters and our CIMP-specific metabolic signature. Based on the classification from the random forest prediction factor analysis (Figure 1F), the Myc epigenetic score from the LoVo ChIP-seq (Appendix A), and the methylation predictive score, the best candidates for discriminating the pathological parameters from our TCGA cohort were GAPDH, TYMS, DHFR, and TK1. All these markers had a specific increased expression in CIMP (Figure 4C, first two left panels) and were correlated to hypermutation and MLH1 silencing (Figure 4C, last two right panels). For further analysis, we built a multivariate deep-learning model for neural networks using keras tensorflow Python libraries and these four Myc-specific targets. We found that this model was very efficient in predicting the hypermutation and MLH1 silencing status of tumours (95 and 96%, respectively) but also the CIMP and MSI status (82 and 85%, respectively; Table 2). Predicting the MSI and MLH1 status is of particular interest, since it is one of the only parameters used to stratify patient care and decide on the use of immunotherapies [32]. Very interestingly, three of our four markers (TYMS, DHFR, and TK1) are also part of the one-carbon metabolism pathway.

### 3.5. Overexpression of One-Carbon Metabolism Enzymes Is An Independent Marker of Methylation Status, MLH1 Silencing, Hypermutation, and MSI in Colorectal Cancer

To study in more detail the prognostic value of one-carbon metabolism in CRC, we selected the enzymes significantly associated with the methylation status (TYMS, TK1, SHMT2, MTHFD1, MTHFD2, and DHFR; Figure 4B), and their normalized Z-scores were transformed into quantile-five (Q5) classes. This allowed us to perform iterative Chi-square tests, with the different clinical parameters (for details see, Appendix A) inputted as character strings for Natural Language Processing (NLP) (see Materials and Methods for more details; Figure 5A). Amongst the one-carbon metabolism genes, the expression of TYMS had the highest association with the hypermutation status, iCluster classification, methylation subtypes, MLH1 silencing, and MSI status (Figure 5A). Consistent with these proteins being part of the same pathway, the same associations, although with less significance, were found for TK1, SHMT2, and MTHFD2 (Figure 5A). MTHFD1 did not show significant association with MSI status and was not able to discriminate between CIN/CIMP (Expression_subtype variable in Figure 5A). DHFR showed the weakest association with clinical parameters and was not associated with methylation or iCluster variables (Figure 5A). Interestingly, none of the one-carbon metabolism genes were able to discriminate for tumour stage (“Tumour stage” variable). This was consistent with the fact that stratifying patients per methylation status gave a similar distribution of tumour stages (Table 1).

We next performed a variable community detection using a Louvain algorithm on the results from the chi-square tests (Figure 5A). This analysis confirmed the association of TYMS, TK1, SHMT2, MTHFD2, and DHFR expressions with the main prognosis markers (hypermutation, MSI, iCluster, expression subtype, and tumour stage) while MTHFD1, and not MTHFD2, was more strongly associated with methylation status (Figure 5B). Based on these observed associations, we finally built a methylation status multivariable model with a minimum of confounding parameters based on the expression of these one-carbon metabolism genes and clinical parameters. This logistic multivariate model particularly confirmed that overexpressed TYMS and MTHFD1 are strong, independent, one-carbon metabolism markers able to discriminate for the CIMP phenotype (Figure 5C). It also confirmed that CIMP tumours are strongly associated with MSI, as previously published [16].

### 3.6. Activation of 1-C Metabolism Genes Predicts Colorectal Cancer Patients with Worst Prognosis

In order to validate the association of one-carbon metabolism and the hypermethylation status as bad prognosis markers for colorectal cancer patients, multi-omics analysis was performed in the 2018 augmented CRC dataset (TCGA-COAD, our validation cohort) from the TCGA consortium [23]. Using the Gene Ontology database, through the AMIGO website, genes from the one-carbon metabolism were selected (Appendix A). We looked at the overexpression of these one-carbon metabolism genes in the CRC tumour RNAseq data and selected the most changed to establish a gene signature (Figure 6A).

Overall survival analysis revealed that patients presenting alterations in this one-carbon signature (i.e., increased expression of the genes in the signature) had worse survival (Figure 6B). As previously reported [2], tumours from the right colon (comprising the ascending and transverse colon, hepatic flexure, and cecum) showed a significant association with increased one-carbon metabolism (Figure 6C) and were associated with a higher age at diagnosis. Looking at HM450 methylation profiles, we found that an altered one-carbon signature was associated with an increase in the global methylation of the genome, consistent with a role for the one-carbon metabolism in providing methyl groups for methylation (see Figure 6D as well as Appendix A for a detailed list of genes with differentially methylated promoters).

We then performed a functional enrichment analysis of the differentially methylated genes using the KEGG or Gene Ontology database (Figure 6E,F). This revealed that most changes were seen in primary bile acid biosynthesis, but also affected vitamin absorption, PPAR signalling, and stem cell functions (WNT signalling, pathways of pluripotency regulation). We also used the Gene Ontology database to perform functional enrichment and found the most changes in genes implicated in the regulation of SMAD protein phosphorylation.

Altogether, these results confirmed that the activation of one-carbon metabolism in CRC is associated with a worse prognosis for patients and a hypermethylation of gene promoters that could impact important functionalities in intestinal cells and have causative effects on cancer progression and dissemination. This strongly confirms that targeting the one-carbon metabolism is of high importance for CRC patients.

## 4. Discussion

Despite the role of KRAS and BRAF in driving the CIMP phenotype and rewiring cancer metabolism [2,33,34,35], few studies have aimed at establishing whether CIMP tumours have a different metabolism [5]. Using transcriptomics data and focusing only on metabolic pathways, we found that CIMP tumours indeed present a hypermetabolism, with surprisingly no down-regulated but several up-regulated metabolic pathways. It is important to note that although hypermethylation leads to down-regulation of genes, only 355 genes out of nearly fifteen thousand tested were hypermethylated in the cohort we used to generate our models [15], indicating that the increase in hypermethylation observed in CIMP tumours is not a pan-genome but rather a very selective event. We hence determined a metabolic signature specific to CIMP tumours, able to discriminate hypermethylated tumours and even separate CIMP-L from CIMP-H patients with high accuracy. A four-gene signature, consisting of TYMS, GAPDH, TK1, and DHFR, was enough to diagnose the CIMP phenotype and identify tumours eligible for immunotherapy (Table 2). Our multivariate model even showed that TYMS and MTHFD1 expressions alone are independent predictors of the methylation status (Figure 5C). This suggests the possibility of identifying these higher-risk tumours with high accuracy, using transcriptomics data, which could provide an alternative to the current techniques using the methylation of a panel of genes, with several gene panels being used over the world and no consensus on which panel and which technique is most suitable for testing for methylation [4]. Furthermore, one-carbon metabolism enzymes were amongst the most efficient predictors of the CIMP phenotype and able to differentiate patients with poorer prognosis, confirming the importance of this pathway, most likely in providing methyl groups to support the high methylation of the genome. Indeed, BRAF mutation on its own is not able to drive carcinogenesis and rather triggers senescence and death [36]. The background of high genome methylation observed in CIMP is hence a pre-requisite for mutant BRAF to be pro-tumorigenic; this is what is thought to foster the development of BRAF-driven CIMP tumours [37].

Myc expression is changed in almost all cases of colorectal cancer, underlying its role in tumorigenesis [16]. Amongst the genes in our metabolic signature, one-third could be under the control of Myc in colorectal patients. Indeed we confirmed, using ChIP-seq data, that Myc physically binds to the promoter of these genes in the LoVo colorectal cancer cell line. Paradoxically, Myc expression is lowest in CIMP-high tumours, compared to the other tumour subtypes, where the signature is observed. We also found that binding sites for Max, one of the best characterized co-factors of Myc whose expression is highest in CIMP-high tumours, are present within the promoters of the metabolic signature, suggesting that the specificity of the signature could be coming from the expression of co-factors rather than of Myc itself. Such a phenomenon has already been observed in Small Cell Lung Cancer, where it has been shown that Max expression regulates the 1C-metabolism pathway in a context-dependent manner [38].

Several meta-analysis of CRC cohorts have now clearly established that the CIMP phenotype is associated with a worse prognosis and that the benefit of an adjuvant fluorouracil (5-FU, one of the classical chemotherapy protocols for CRC patients) treatment after surgery is limited for CIMP-high patients, especially those with advanced stage III–IV tumours [3,39,40]. Given the important toxicity of these treatments, identifying those patients is very important in order to improve their quality of life and prevent their exposure to a medication with limited interest and high toxicity for them. We found that one of the genes most associated and predictive of the CIMP phenotype is TYMS (thymidylate synthase), which has been shown to be up-regulated by Myc [41] and to drive resistance to 5-FU treatments, although it is the target of the drug [42,43]. Hence, since the 1C-metabolism pathway seems to play a central role in CIMP tumours, other targets in the pathway should be explored. Indeed, our results suggest that MTHFD1, 2 could be very interesting targets for the treatment of colorectal cancer. MTHFD2 is of particular interest, since it has been shown to be expressed only in embryonic tissues but becomes re-activated in cancer tissues [44,45]. It is hence a tumour-specific marker that has been shown to be overexpressed and associated with a worse prognosis in many cancer tissues [45,46,47]. It has also been recently proposed to play a role in the regulation of the immune system, fostering cancer immune evasion [48]. Interestingly, in this study, the authors showed that MTHFD2 promotes O-glycosylation, which increases Myc’s stability and PD-L1 transcription. A recent study showed that inhibitors of MTHFD2 lead to the specific death of MTHFD2-expressing cells, suggesting that these therapeutic approaches could be very specific to cancer cells and have minimal impact on normal cells.

## 5. Conclusions

Our study shows that transcriptomics holds great power in unbiasedly identifying hypermethylated tumours. This approach could provide an alternative to the actual disparities in the methods used to assess methylation and allow more consistency and robustness. Additionally, we show that transcriptomics identified tumours with high MSI and MLH1 silencing, which are indications of good responders to immunotherapy, and these patients should receive this treatment rather than 5-FU [32]. This is in accordance with recent studies showing that transcriptomics is becoming more and more relevant in the clinical practice and provides a great clinical benefit to orient treatment protocols, especially for patients without tractable DNA mutations in their tumours [49].

## Figures and Tables

**Figure 1 biomedicines-12-00590-f001:**
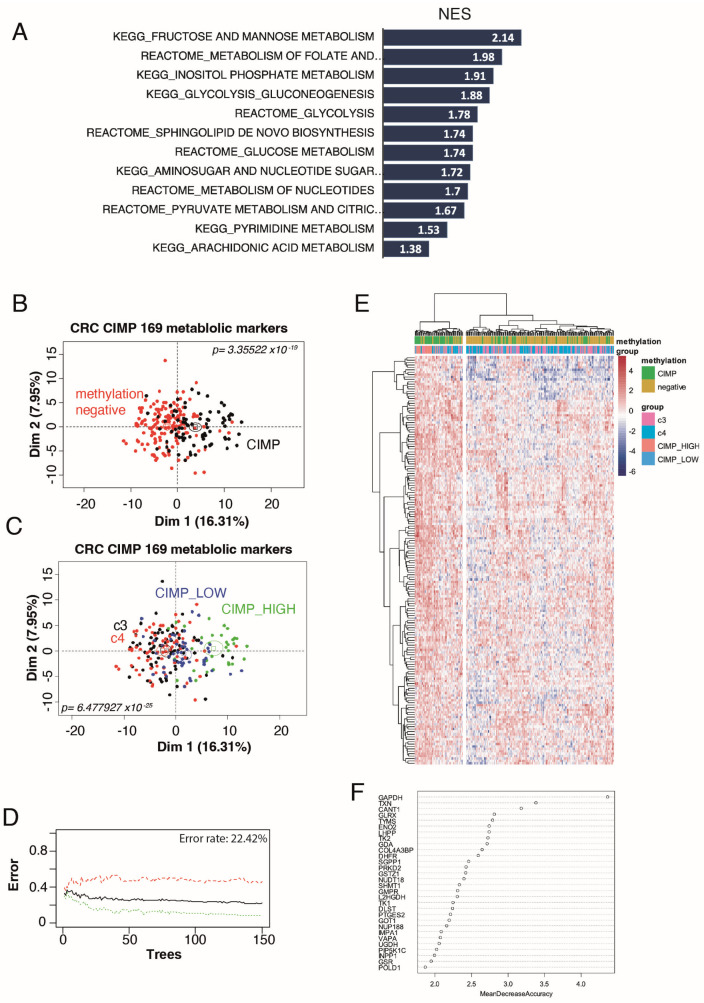
Hypermetabolism in CIMP CRC transcriptome. (**A**) Barplot of metabolic pathway found over-expressed in CIMP CRC as compared to unmethylated ones (NES: normalized enrichment scores); (**B**) unsupervised PCA on CRC transcriptome separates unmethylated and CIMP CRCs using the 169 metabolic marker signature; (**C**) unsupervised PCA on CRC transcriptome can stratify tumours into four clusters—C3, C4, CIMP-low, and CIMP-high—using the 169 metabolic marker signature; (**D**) machine learning misclassification error rate estimation by random forest analysis to classify methylation status using the metabolic expression profile (mismatch classification error curves for CIMP+ patients (green), CIMP- patients (red) and the mean of the two groups (black)); (**E**) unsupervised classification of CRC tumours with the metabolic signature (Euclidean distance and Ward.D2 method); (**F**) variable importance plot of the best metabolic markers to stratify by methylation status (clusters c3 and c4, CIMP_low, and CIMP_high).

**Figure 2 biomedicines-12-00590-f002:**
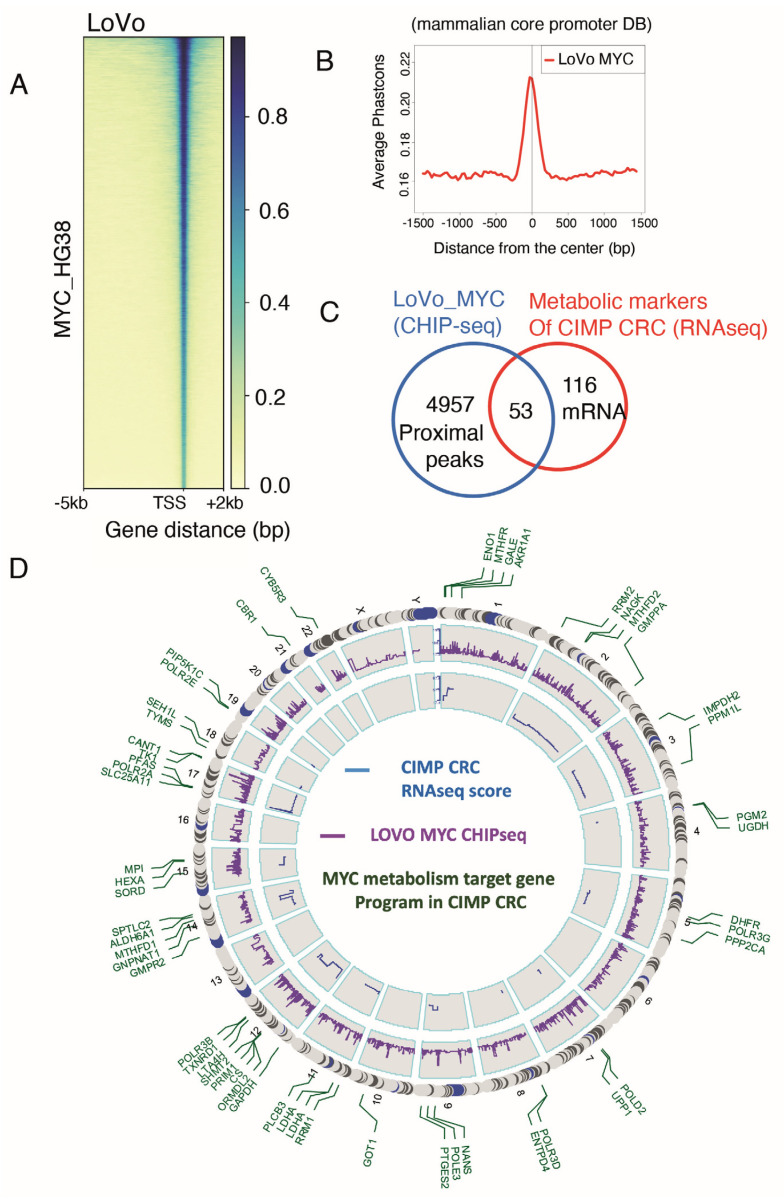
Myc targets in CIMP CRC metabolism activated program. (**A**) Promoter heatmap for Myc chromatin binding in the whole genome of the LOVO colorectal cell line (matrix computed on proximal promoter regions: minus 5 kb and plus 2 kb around TSS); (**B**) phast conservation plot of LOVO Myc CHIP-seq around mammalian core promoter database; (**C**) Venn diagram showing genes at the intersection between the LoVo Myc CHIP-seq dataset and the metabolic signature in CIMP; (**D**) circus plot presenting the results of the Myc CHIP-seq epigenetics analysis (purple: peak scores) and the mRNA levels in the transcriptome of CIMP CRC (blue: delta between CIMP and unmethylated CRC). Genes present in the metabolic signature are indicated in green. HG19 was used as the referent genome.

**Figure 3 biomedicines-12-00590-f003:**
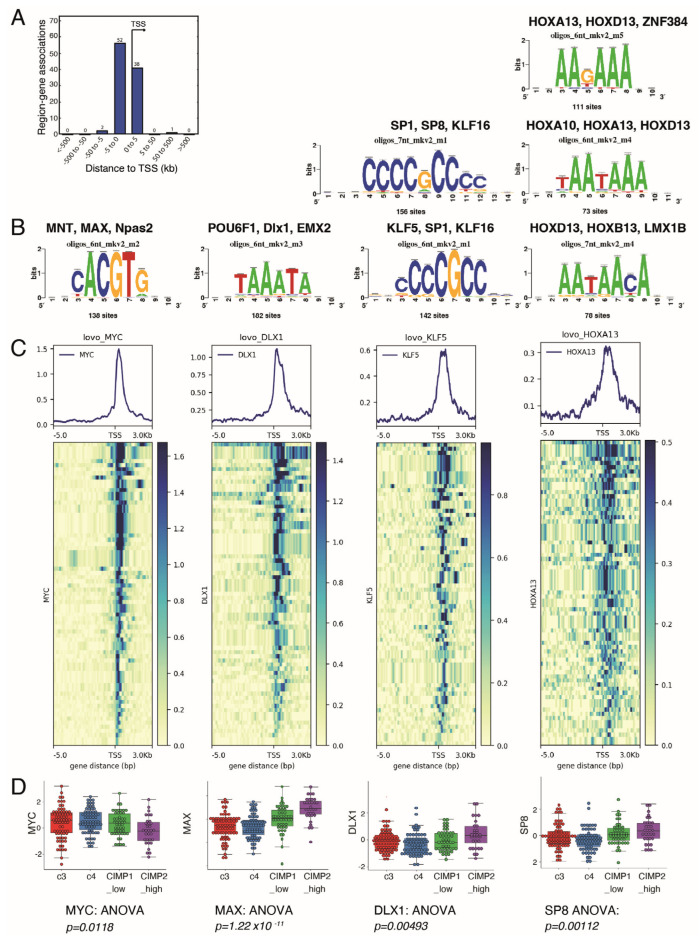
Myc-driven hypermetabolism gene signature presents binding sites for transcription factors. (**A**) Promoter barplot of all the Myc targets in the CIMP metabolic signature; (**B**) motif prediction for binding on Myc intervals in the promoter of genes from the metabolic signature, based on Jaspar vertebrae 2018 motif prediction database; (**C**) promoter heatmaps using CHIP-seq data from LoVo cells for Myc, DLX1, KLF5, and HOXA13 binding on the promoters of the 169 genes found in the CIMP metabolic signature; (**D**) expression boxplot of Myc, Max, Dlx1, and SP8 transcription factors in the transcriptome of CRC patients, stratified by methylation status.

**Figure 4 biomedicines-12-00590-f004:**
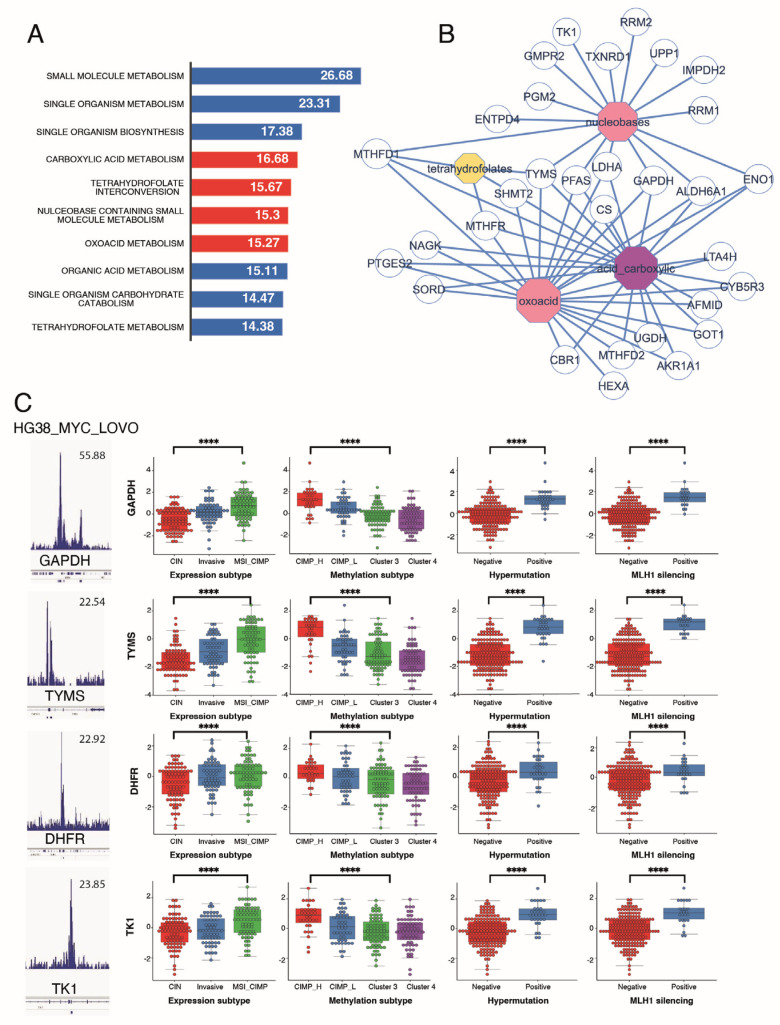
Genes in the CIMP-specific metabolic signature are associated with worse clinical outcomes. (**A**) Functional enrichment (Biological Process of Gene Ontology database: GO-BP) performed on Myc metabolic targets overexpressed in the CIMP signature. Barplots represent logarithm 10 of negative binomial *p*-values; (**B**) Functional enrichment performed on the main metabolic functions enriched for Myc targets from the CIMP signature over-expressed in CIMP-CRC; (**C**) Selection of four discriminant metabolic Myc targets dysregulated in CIMP-CRC: for each, the Myc LOVO CHIP-seq and expression boxplots as stratified according to clinical data (methylation subgroups, expression subgroups, hypermutation status, and MLH1 silencing) are represented. **** *p* < 0.0001.

**Figure 5 biomedicines-12-00590-f005:**
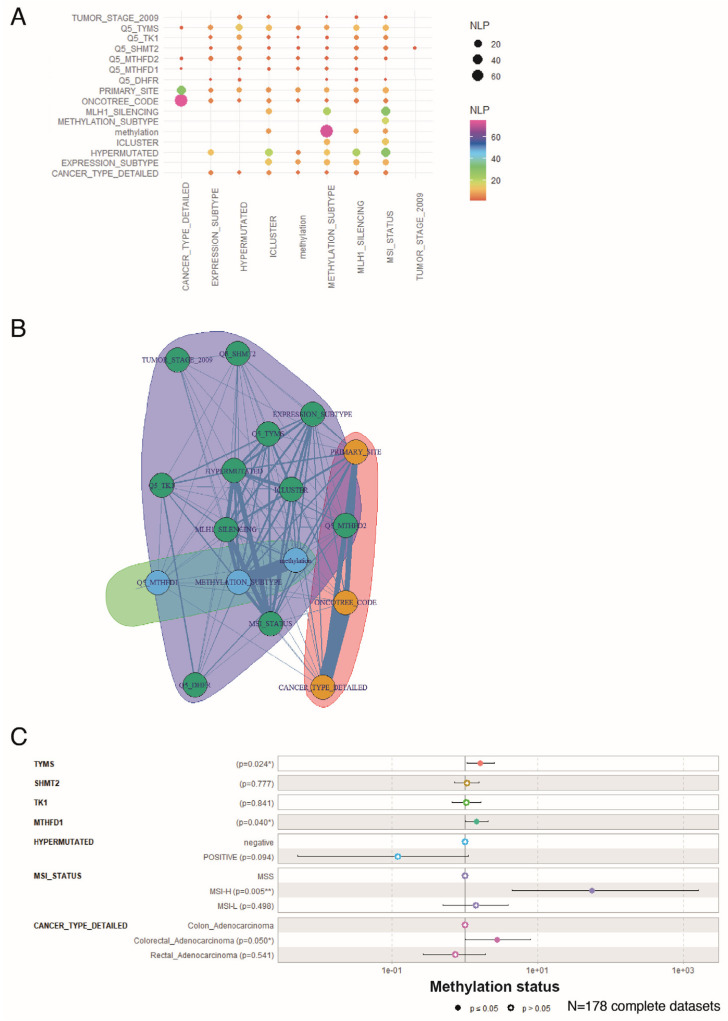
One-carbon metabolism multivariable model stratified by methylation status. (**A**) Negative log10 *p*-values (NLPs) of iterative chi-square tests performed between parameters; (**B**) network established for the detection of communities between parameters; (**C**) multivariable model showing the association of one-carbon metabolism gene expression and clinical parameters with tumour methylation status, tested as logistic regression. * *p* < 0.05; ** *p* < 0.01.

**Figure 6 biomedicines-12-00590-f006:**
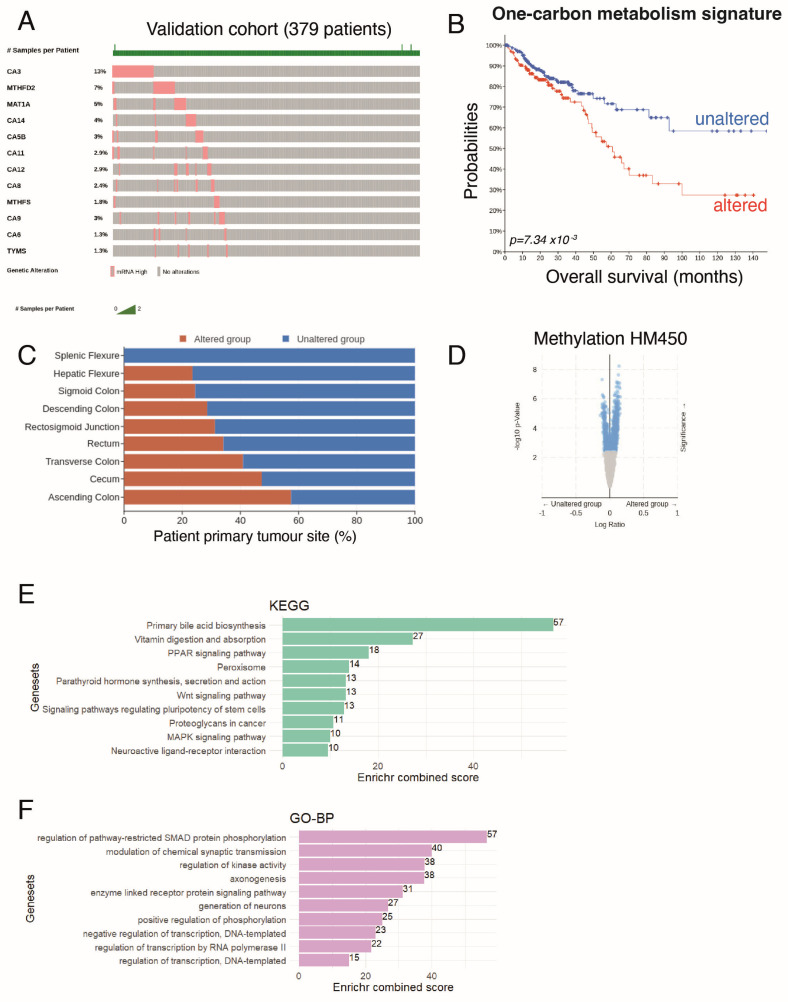
Assessment of one-carbon metabolism activation with regard to overall survival prognosis in colorectal cancer. (**A**) Oncoprint analysis of one-carbon metabolism gene candidates found to be overexpressed in CRC tumours; (**B**) Kaplan–Meier and log-rank test performed between altered and unaltered groups of CRC patients focused on one-carbon metabolism overexpression in RNA-sequencing; (**C**) barplot of tumour localization stratified by carbon metabolism alteration status; (**D**) volcano plot for differential methylation HM450 quantification between altered and unaltered one-carbon metabolism CRC patients; (**E**) KEGG (Kyoto Encyclopedia of Genes and Genomes) functional enrichment performed on hypermethylated gene promoters of patients presenting overexpression of one-carbon metabolism genes in their transcriptomes; (**F**) gene-ontology biological process enrichment performed on hypermethylated gene promoters of patients presenting overexpression of one-carbon metabolism genes in their transcriptomes.

**Table 1 biomedicines-12-00590-t001:** Clinical information of colorectal cancer RNA-seq cohort stratified by methylation status: Univariate logistic regression was performed on the available clinical parameters present in the TCGA CRC cohort (2012 [16]) and stratified by methylation status (referred to as “Positive” or “Negative” in columns 2 and 3 of the table).

Variable	Subtypes	Negative (n = 143)	Positive (n = 80)	Total (n = 223)	*p*-Value
MSI_STATUS	MSS	116 (81.1)	41 (51.9)	157 (70.7)	
	MSI-L	24 (16.8)	13 (16.5)	37 (16.7)	
	MSI-H	3 (2.1)	25 (31.6)	28 (12.6)	<1 × 10^−4^
	missing	0	1	1	
METHYLATION_SUBTYPE	Cluster3	74 (51.7)	0 (0.0)	74 (33.2)	
	Cluster4	69 (48.3)	0 (0.0)	69 (30.9)	
	CIMP_H	0 (0.0)	32 (40.0)	32 (14.3)	
	CIMP_L	0 (0.0)	48 (60.0)	48 (21.5)	<1 × 10^−4^
ICLUSTER	c1	43 (36.8)	11 (16.7)	54 (29.5)	
	c2b	16 (13.7)	22 (33.3)	38 (20.8)	
	c3	48 (41.0)	9 (13.6)	57 (31.1)	
	c2a	10 (8.5)	24 (36.4)	34 (18.6)	<1 × 10^−4^
	missing	26	14	40	
MLH1_SILENCING	negative	142 (99.3)	56 (70.0)	198 (88.8)	
	positive	1 (0.7)	24 (30.0)	25 (11.2)	<1 × 10^−4^
EXPRESSION_SUBTYPE	CIN	77 (54.6)	11 (13.9)	88 (40.0)	
	Invasive	36 (25.5)	25 (31.6)	61 (27.7)	
	MSI_CIMP	28 (19.9)	43 (54.4)	71 (32.3)	<1 × 10^−4^
	missing	2	1	3	
HYPERMUTATED	negative	125 (93.3)	51 (69.9)	176 (85.0)	
	positive	9 (6.7)	22 (30.1)	31 (15.0)	<1 × 10^−4^
	missing	9	7	16	
CANCER_TYPE	Colorectal_Adenocarcinoma	143 (100)	80 (100)	223 (100)	<1 × 10^−4^
CANCER_TYPE_DETAILED	Colon_Adenocarcinoma	84 (58.7)	43 (53.8)	127 (57.0)	
	Colorectal_Adenocarcinoma	15 (10.5)	23 (28.8)	38 (17.0)	
	Rectal_Adenocarcinoma	44 (30.8)	14 (17.5)	58 (26.0)	0.001041
ONCOTREE_CODE	COAD	84 (58.7)	43 (53.8)	127 (57.0)	
	COAD-READ	15 (10.5)	23 (28.8)	38 (17.0)	
	READ	44 (30.8)	14 (17.5)	58 (26.0)	0.001041
PRIMARY_SITE	3-left colon	59 (41.5)	13 (16.2)	72 (32.4)	
	1-right colon	24 (16.9)	42 (52.5)	66 (29.7)	
	2-transverse colon	5 (3.5)	9 (11.2)	14 (6.3)	
	4-rectum	54 (38.0)	16 (20.0)	70 (31.5)	<1 × 10^−4^
	missing	1	0	1	
TUMOR_STAGE_2009	Stage_IIA	46 (32.6)	33 (41.8)	79 (35.9)	
	Stage_IIIC	17 (12.1)	3 (3.8)	20 (9.1)	
	Stage_IIIB	20 (14.2)	11 (13.9)	31 (14.1)	
	Stage_I	31 (22.0)	15 (19.0)	46 (20.9)	
	Stage_IIIA	3 (2.1)	1 (1.3)	4 (1.8)	
	Stage_IV	22 (15.6)	12 (15.2)	34 (15.5)	
	Stage_IIB	2 (1.4)	3 (3.8)	5 (2.3)	
	Stage_IVA	0 (0.0)	1 (1.3)	1 (0.5)	0.29398
	missing	2	1	3	

**Table 2 biomedicines-12-00590-t002:** Four MYC metabolic targets were used for deep learning predictions of prognostic status in patients with colorectal cancer. The neural network is based on the RNA quantification of 4 Myc metabolic targets, GAPDH + TYMS + TK1 + DHFR, to predict each prognostic parameter.

CRC Status	Number of Patients	Accuracy	Precision	Recall	F1 Score	Cohen Kappa Score	AUC: Area under Curve
methylation CIMP	223	0.82	0.78	0.72	0.75	0.62	0.90
Hypermutation	207	0.95	0.87	0.83	0.85	0.82	0.98
MLH1 silencing	223	0.96	0.86	0.80	0.83	0.81	0.99
MSI	222	0.85	0.83	0.63	0.72	0.62	0.94

## Data Availability

Links to all algorithms used in this study are listed in the Materials and Methods section.

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
