# Peer review of "Hypermethylated Colorectal Cancer Tumours Present a Myc-Driven Hypermetabolism with a One-Carbon Signature Associated with Worsen Prognosis"

_biomedicines, 2024, doi:10.3390/biomedicines12030590_

Round 1
Reviewer 1 Report
Comments and Suggestions for Authors
The aim of this manuscript was to test whether the methylation status of colorectal cancer is associated with metabolic alterations. The authors performed an extensive analysis of RNA-sequencing data available online using several machine learning and deep learning methods. They found that colorectal cancer with a high activation of the CpG-island methylation pathways (CIMP- high) showed an upregulation of many metabolic pathways including carbohydrates, folates, sphingolipids and arachidonic acid biochemical pathways. The metabolic signature comprising 169 enzymes was able to discriminate methylated and unmethylated colon cancers. Furthermore, by using online available data from ChIP sequencing experiments in the colorectal cell line LoVo, they found that one third of the involved genes were regulated by MYC and its co-factor MAX.
The study is interesting and contains many information that may prove useful in a clinical context. However, the reported findings were not surprising since the CIMP phenotype is associated with KRAS and BRAF mutations and it is well-known that activation of growth factor receptor signaling cascade leads to many metabolic alterations through its effectors MYC/MAX. Therefore, the association of CIMP phenotype with a hypermetabolic state can be due to the presence of KRAS/BRAF mutations. On the other hand, it is counterintuitive that the CIMP phenotype characterized by repression of genes is associated with upregulation of all metabolic pathways unless there is evidence of repression of multiple metabolic regulatory genes. The authors should better explain in the discussion section the interplay between CIMP and KRAS/BRAF mutations.
Legend to Table 1 should be more explicative. The term negative and positive for the column 3 and 4 refers to the presence or absence of metylation status. This should be indicated in the table. Also, the term “level” for the second column appears to be inappropriate. What about subtypes?
In the result section the CRC-2012 TCGA cohort comprises 222 patients whereas in the materials and methods section and in the table 1 it includes 223 patients. Please, clarify.
Please, do not refer to methylated and unmethylated patients but to methylated and unmethylated colorectal cancers or samples throughout the manuscript including the title.
English should be carefully revised
Comments on the Quality of English LanguageEnglish should be carefully revised
Reviewer 2 Report
Comments and Suggestions for Authors
This is a largely bioinformatic study analysing different subgroups of CRC. The study could be useful in improving diagnosis and treatment of CRC. My comments are largely for clarification.
Line 53: Please clarify, methylation can occur in CpG islands in promoters and also in amino acid residues of histones. The CIMP phenotype refers to methylation of CpG islands not histones.
Line 57: Is an increase of or decrease of one carbon metabolism associated with poor outcomes. The metabolism itself cannot be associated with an outcome.
Line 177: Given that the authors have emphasized the need to specify the sites of methylation, this information should be provided here.
Line 182: After emphasizing in the introduction that methylation is associated with down-regulation of genes, this deserves discussion, which the authors avoid apart from "surprising".
Line 224: cMyc is but one of the targets of BRAF signaling. Why were other transcription factors excluded?
Line 247: I assume this should be base pairs: bp
Line 408-411: The authors could discuss their preferred panel here or the transcriptomic analysis of which specific genes.
Line 422: This supports my comments made above on the non-exclusive role of cMyc.
Line 432: if these tumors are so reliant on C1 metabolism shouldn't 5-FU be beneficial as a treatment?
Round 2
Reviewer 1 Report
Comments and Suggestions for Authors
The authors have answered all my questions and revised the manuscript accordingly.